# Children’s Pain Identification Based on Skin Potential Signal

**DOI:** 10.3390/s23156815

**Published:** 2023-07-31

**Authors:** Yubo Li, Jiadong He, Cangcang Fu, Ke Jiang, Junjie Cao, Bing Wei, Xiaozhi Wang, Jikui Luo, Weize Xu, Jihua Zhu

**Affiliations:** 1College of Information Science and Electronic Engineering, Zhejiang University, Hangzhou 310027, China; 22231063@zju.edu.cn (J.H.); 21960347@zju.edu.cn (K.J.); cao_jj@zju.edu.cn (J.C.); xw224@zju.edu.cn (X.W.); jackluo@zju.edu.cn (J.L.); 2International Joint Innovation Center, Zhejiang University, Haining 314400, China; 3Children’s Hospital, Zhejiang University School of Medicine, Hangzhou 310052, China; fucangcang@zju.edu.cn (C.F.); weizexu@zju.edu.cn (W.X.); 4Polytechnic Institute of Zhejiang University, Hangzhou 310015, China; sgweibing@zju.edu.cn; 5National Clinical Research Center for Child Health, Hangzhou 310052, China

**Keywords:** skin potential (SP), children’s pain identification, machine learning, feature extract

## Abstract

Pain management is a crucial concern in medicine, particularly in the case of children who may struggle to effectively communicate their pain. Despite the longstanding reliance on various assessment scales by medical professionals, these tools have shown limitations and subjectivity. In this paper, we present a pain assessment scheme based on skin potential signals, aiming to convert subjective pain into objective indicators for pain identification using machine learning methods. We have designed and implemented a portable non-invasive measurement device to measure skin potential signals and conducted experiments involving 623 subjects. From the experimental data, we selected 358 valid records, which were then divided into 218 silent samples and 262 pain samples. A total of 38 features were extracted from each sample, with seven features displaying superior performance in pain identification. Employing three classification algorithms, we found that the random forest algorithm achieved the highest accuracy, reaching 70.63%. While this identification rate shows promise for clinical applications, it is important to note that our results differ from state-of-the-art research, which achieved a recognition rate of 81.5%. This discrepancy arises from the fact that our pain stimuli were induced by clinical operations, making it challenging to precisely control the stimulus intensity when compared to electrical or thermal stimuli. Despite this limitation, our pain assessment scheme demonstrates significant potential in providing objective pain identification in clinical settings. Further research and refinement of the proposed approach may lead to even more accurate and reliable pain management techniques in the future.

## 1. Introduction

### 1.1. Background

Pain is a complex physiological and psychological activity which is one of the most common symptoms in clinical practice [1]. It can result from actual or impending tissue damage or psychological factors [2]. Pain is considered the fifth vital sign after temperature, pulse, respiration, and blood pressure. Studies have shown that repeated or intense pain stimulation in children can lead to hormonal disruptions that may persist from childhood to adulthood [3]. Therefore, pain management for children is crucial. However, inadequate pain management can increase the length of stay and overall healthcare costs [4]. However, since a significant characteristic of pain is a tremendous individual difference. The subjective nature of pain makes it difficult to manage, as it lacks an objective evaluation basis, unlike the other four vital signs [5,6].

There are three standard methods for pain identification: traditional pain assessment scales, such as the Neonatal Facial Coding System (NFCS), CRIES Scale, FLACC Scale, Children’s Hospital of Eastern Ontario Pain Scale (CHEOPS), Visual Analogue Scale (VAS), and Wong–Baker Facial Expression Scale [6,7,8,9]; identifying pain intensity from behavioral cues (e.g., facial expressions) [10,11,12,13]; and identifying pain intensity by physiological signals, such as skin conductance (SC), heart rate (HR), photoplethysmography (PPG), electromyography (EMG), electrocardiogram (ECG), and electroencephalogram (EEG) [1,14,15,16,17,18,19].

Various factors make it challenging for children to accurately describe their pain, such as their cognitive abilities, behavioral responses, and emotional expressions that differ with age [20,21]. Additionally, the scales cannot continuously assess pain in children. Likewise, EEG and ECG are unsuitable for children due to the bulky measurement equipment. While PPG is indeed a portable and reliable measurement device [22], it relies on optical principles for measurement. This paper is concerned with the electric method in our experiment. Therefore, we have opted to use a skin-potential-based approach to pain identification. This electrical approach allows for a more feasible and non-invasive pain assessment and is better suited for our research objectives. We acknowledge that optical methods have their merits, but our focus in this study was to explore an electrical approach to pain identification in pediatric patients.

### 1.2. Related Works

Existing studies have mainly focused on pain induced by heat, electricity, or laser, with pain intensity assessed by bioelectrical signals such as SC, ECG, and EEG. Although these research methods ensure consistent pain stimulation, the pain production in these studies is not the same as clinical pain. Therefore, they cannot be directly applied to detect clinical operative pain.

Daniel et al. [14] used data from the BioVid heat pain database, which includes SC and ECG. The pain was classified into four levels using logistic regression (LR), support vector machine (SVM), single-task neural network (STNN), and multi-task neural network (MTNN) algorithms. The accuracy of identification for the highest pain level reached 82.75%. Kong et al. [15] used a wrist-worn device to collect electrical skin activity (EDA). The pain was evoked by using both thermal and electrical stimulation, and an accuracy of 81.5% was obtained in pain detection using a random forest algorithm. However, pain induced by thermal or electrical stimulation does not represent the actual pain condition of clinical patients and cannot be directly translated to clinical applications.

Treister et al. [16] produced different pain levels by stimulating 45 healthy volunteers and recorded photoplethysmography (PPG), galvanic skin response (GSR), and ECG. The results showed that these physiological signals differed significantly between all pain levels. Ben-Israel Nir et al. [17] studied indices of injury perception levels in patients under general anesthesia. The combined use of the physiological parameters HR, HRV, PPG, and GSR was better than any individual physiological signal in assessing the injurious response. Both methods obtained good multiparametric analyses, but they required assembling many types of equipment and were difficult to operate clinically.

Schulz E et al. [18] used a laser to apply pain stimuli to healthy volunteers and recorded the subjects’ brain activity by EEG. They analyzed the data using multivariate analysis methods and cross-validation and found that a classifier trained on a group of healthy individuals could predict pain sensitivity in another individual with a prediction accuracy of 83%.

This paper identified a physiological signal called the skin potential (SP) as a means of identifying pain. SP has been associated with emotional changes since the 1880s. Skin potential and skin conductance represent different manifestations of the same neurophysiological systems and are correlated [23]. However, because SP is difficult to obtain and analyze, it is often ignored in recent research.

## 2. Materials and Methods

### 2.1. Participants

We randomly selected 623 subjects to participate in our clinical trial. All participants were children whose ages ranged from 2 to 18 years. The sex statistics and age distribution of the participants are shown in Figure 1.

### 2.2. SP Characteristics of Pain

In our experiment, we used the middle finger and the inner wrist as the measurement and reference points to localize the pain signal. These two points were chosen based on our extensive prior experimental experience.

Figure 2a shows a typical SP signal for pain in a subject. The position shown by the red arrow in Figure 2a is the needling time. Intuitively, the signal is characterized by a rapid change process of rising and falling after the pain is triggered by the needling. Figure 2a shows that the signal fluctuates between −33 and −27 mV. However, the specific fluctuation range varies depending on the individual state of the subject, including the signal pattern. As shown in Figure 2b, the signal pattern is changed due to emotional fluctuations such as crying. Therefore, the rapid rise and fall cannot be identified as the SP signal corresponding to pain, as described below. Figure 2c shows the results of extracting the one-sided spectrum after performing the fast Fourier transform (FFT) on the signal in Figure 2a. It can be seen that most of the energy of the SP signal is concentrated at very low frequencies (within 0.5 Hz), and the energy of its DC segment is relatively high. Therefore, we extracted and performed AI identification of pain for SP signals in the range of 0~0.5 Hz.

### 2.3. Experiment

Figure 3 shows our pain stimulation experiment. First, we attached the device to the subject’s hand, and the specific position of our device was the middle finger and hand. Then, we asked the subjects to keep silent for more than 30 s to establish a baseline of the dynamic characteristics of the subject’s SP signal without external stimulation. Then, we performed clinical venipuncture on the subject’s other hand, as shown in Figure 3. During this process, the instrument measured the SP signal and accurately recorded the puncture time by the self-developed APP. With all the information collected here, the extraction and analysis of clinical operative pain SP features were performed to establish an AI model for pain detection.

### 2.4. Preprocessing

#### 2.4.1. Data Cleaning

To ensure the reliability of the algorithm results, we needed to perform data cleaning. Because the subjects were all children, we found that many subjects had difficulty maintaining silence as required during the experiment. Children may cry loudly before the intravenous blood draw due to fear, pain, and other factors. Such emotional changes resulted in significant, high-frequency changes in the stability of the measured SP signal before the onset of operant pain, as shown in Figure 2b. This type of signal resulted in our inability to establish the dynamic characteristic baseline.

To exclude the interference of subjects’ crying with the experimental results, we added a process to remove crying interference. We observed and recorded the subjects’ crying in detail during the experiment. Subsequently, the recorded data from crying subjects were excluded from the analysis, ensuring a clean dataset that adheres to the experimental criteria for pain feature extraction. Therefore, the data sample was reduced from 623 cases to 358 cases.

#### 2.4.2. Normalization

The following equation normalized the data from the above 358 experiments for each subject’s SP data.
$$V_{norm}=(V-V_{min})/(V_{max}-V_{min})$$

*V* represents the measured voltage value, $V_{min}$ represents the minimum value of the measured voltage, $V_{max}$ represents the maximum value of the measured voltage, and $V_{norm}$ represents the final normalized result.

#### 2.4.3. Data Slice

Due to the different lengths of the data samples, pre-processing is needed to facilitate machine learning. In much of the related research, samples are typically divided into segments of consistent lengths. Consequently, we have also adopted this approach and partitioned the data into equal-length segments, as demonstrated below [24,25].

During the blood collection operations, we found that the period was between 15 and 25 s. Therefore, the length of the data slices was set to 15 s uniformly.The silent time was more than 30 s, and the silent slice was discarded if there were less than 30 s of experimental data.The starting point of silent samples was calculated from the 10th second after wearing the device. We could exclude the effect of SP signal instability when the device was first put on.If the blood collection operation time was more than 15 s, the starting point of the pain sample started from the moment of recording the needle ligation. Otherwise, we discarded the pain sample slice and kept the silent sample slice before the operation.

A flag bit (tag) was added at the end of the data slice to identify the type of this slice. In this paper, tag 0 was used to mark the silence, and tag 1 was used to mark the pain. The first 60 bits of data are SP signals acquired at 4 Hz, and 1-bit data are the tag field.

### 2.5. Pain Feature Extraction

During the experiment, we found differences in the features of the pain and silent states. The normalized results are shown in Figure 4. Specifically, there is a distinct characteristic peak in the pain sample, the amplitude of the peak is more significant, and the rate of rise and fall is faster than that of the silent sample. Although fluctuations may occur in the silent samples due to various factors, the amplitude of the peak in the silent state is generally significantly smaller than that in the pain state. These features help us to design algorithms for pain identification.

Based on relevant research, we analyzed a total of 38 features, including various statistical features in the time and frequency domains [26,27,28], and found that seven had better pain identification results. We counted the mean and standard deviation of these seven features in all samples, as shown in Table 1.

Notably, standard deviation and variance exhibit a strong correlation, and when either of these features is excluded, there is no significant impact on the model’s recognition rate. Hence, no special treatment is necessary for these features in this context.

We found that the DC segment of the SP signal was the strongest in our study. However, there were individual differences in the DC segment, and finding apparent correlations with external stimuli was challenging. On the other hand, the AC segment of the SP signal was more obviously correlated with external stimuli. Therefore, the seven features shown in Table 1 were all associated with fluctuations. For example, the standard deviation and variance respond to the magnitude of fluctuations in the data mentioned above, and the first difference responds to the rate of change in the data.

### 2.6. Datasets

After pain feature extraction, we obtained 480 sets of sliced samples, of which 262 cases were tagged as 1, and 218 cases were tagged as 0. In order to avoid class imbalance [29], 320 cases were selected as a training set and the rest as a test set. The specific dataset division is shown in Table 2.

### 2.7. Algorithm

To find a better algorithm, we compared the accuracy of three algorithms to classify the dataset, including the K-nearest neighbors (KNN) [30], the random forest (RF) [30,31], and the neural networks (NN) [32]. All of these algorithms were implemented through Python sklearn library.

KNN algorithm finds K points in the feature space closest to the point to be classified. Then it calculates the category of these K points they belong to. The category with the most significant percentage is used as the sample category to be classified. We used cross-validation to determine the value of K. The final value of K was selected as 9.

Random forest is a further optimization of a decision tree, which uses several different decision trees to avoid overfitting anomalous samples. In this study, we set the number of decision trees in the random forest to 1000.

Neural networks are a common classification algorithm used in recent years. In our study, two fully connected hidden layers were used to implement the classification task in the complex plane. The activation function of the neural network model is the relu function.

## 3. Results

### 3.1. Accuracy of Pain Identifying

The final accuracy results for different algorithms and the different number of features are shown in Table 3. We used the “SelectBest” function in the sklearn library for feature selection to obtain the results for the different numbers of features. From the results of the three algorithms, we can see that the accuracy of the KNN algorithm is generally not high, and the random forest has the highest accuracy.

As can be seen from the calculation results with different number of features, all three algorithms reach the highest accuracy when the number of features is 7. As the number of features increases (up to 38), there is an apparent decrease in the accuracy of all algorithms. This shows that among all the features those we introduced in Section 2.5 contribute the most to the accuracy. Finally, we chose the RF algorithm to obtain the highest accuracy 70.63% under the condition that features are identified.

### 3.2. Effect of Sex and Age on Pain Identification

We also evaluated the effect of sex and age on the pain identification accuracy. Table 4 presents the impact of sex on pain identification accuracy. Our test set consisted of 160 samples, with 73 female subjects and 87 male subjects. The accuracy rate for female subjects was 72.60%, while the accuracy rate for male subjects was 70.11%. The results indicate that sex has no significant effect on pain identification accuracy.

Figure 5a shows the age distribution in the test set, where most of the ages have a sample size of around 10. However, only the ages of 2, 3, and 14 have relatively small samples. Figure 5b shows the impact of age on the accuracy of pain identification. As observed, the accuracy of the samples with subjects aged 4–6 years is between 60% and 70%, while the accuracy of the samples with subjects aged 7 years and above is mostly between 70% and 80%. Only the sample with subjects aged 11 years has a lower accuracy of 62.50%. Overall, the pain identification for older subjects was significantly higher than that of younger subjects. This is likely because older subjects are generally more compliant, cooperative, and less likely to cry during the experiments. Additionally, it is possible that older subjects have a more sensitive perception of pain. The specific reasons still need further research.

## 4. Conclusions and Discussion

Our study aimed to investigate the correlation between SP signals and clinical operant pain, focusing on the detection and identification of pain during venipuncture in a clinical setting. Previous studies on pain identification have relied on methods such as electrical, thermal, or laser stimulation to induce pain which may not be directly applicable to clinical operant pain detection. In this paper, we employed a portable SP signal measurement device and designed an experimental paradigm to identify pain during venipuncture.

Compared to other physiological signals such as ECG and EEG used in previous studies, the SP signals do not require complex and large measurement devices, making them easier to measure. Additionally, the use of electrical current during SC measurement can irritate human tissue, potentially reducing patient compliance and affecting measurement results. In contrast, our technique does not require electric current during the measurement process, making it more suitable for clinical application.

In this study, we focused on the extraction of pain features and analyzed the accuracy of different algorithms based on time and frequency eigenvalues. Using the SP signal and RF algorithm, we achieved a final accuracy of 70.63% for operant pain detection, which has practical clinical application. It is worth noting that the level of pain was not analyzed in this paper due to the subjective nature of pain perception. Since venipuncture is a manual procedure, different nurses have different manipulation methods, making it challenging to ensure consistent operant pain triggering. In Table 1, we can see that the standard deviation of the characteristic parameters of the pain samples was more significant than that of the silent samples. However, we collected a large amount of experimental data and used AI identification to reduce the effect of manual manipulation on the results to some extent.

We found that behaviors such as crying can significantly interfere with the experimental results. When crying samples were not excluded, the accuracy of identifying silent and pain states dropped significantly, reaching only 56.64% on the test set using the RF algorithm with the same methodology. In the confusion matrix on the test set, the accuracy of identifying the silent state was only 61.84%, and the accuracy of identifying the pain state was only 50.75%. These results underscore the substantial impact of crying on pain identification, rendering such results unsuitable for clinical application.

As mentioned in Section 2.4.1, the stability of the subject’s SP signal changes substantially and at a high frequency during crying, which is similar to the subject’s SP signal during applied pain stimulation, but since the subject is not subjected to pain stimulation at this time, according to the method described in Section 2.4.3, the algorithm still sets the label field of these data to 0, i.e., the silent state, which will result in many pain samples in the test set with features that may be similar to those of this crying time, being incorrectly identified as the silent state, ultimately leading to a decrease in accuracy. Therefore, in order to obtain a higher accuracy, data with behavioral interference of abnormal emotions need to be excluded from data cleaning.

Overall, the pain identification method based on SP signals is simple and easy to implement, with lightweight and portable measurement equipment that is nonstimulating to the body and has high patient compliance. This technology can help to monitor the physiological electrical indicators of pain in children in real-time, both in home and hospital settings, and can be easily combined with cloud technology to achieve real-time online analysis of dynamic characteristics, wireless technology for children’s pain management, technical support for digital diagnosis and treatment of children’s pain. Compared with traditional assessment scales, our method provides objective and intelligent assessment of children’s pain. In the future, we will continue to explore the detection of pain levels based on SP signals. This research has the potential to provide technical support for digital diagnosis and treatment of children’s pain and help to realize wireless technology for pain management in children.

## Figures and Tables

**Figure 1 sensors-23-06815-f001:**
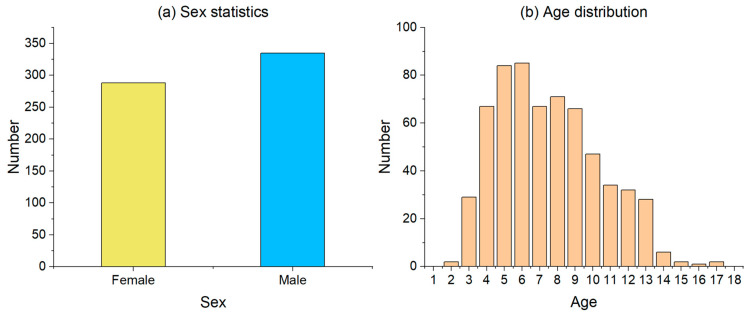
The sex and age information of the participants.

**Figure 2 sensors-23-06815-f002:**
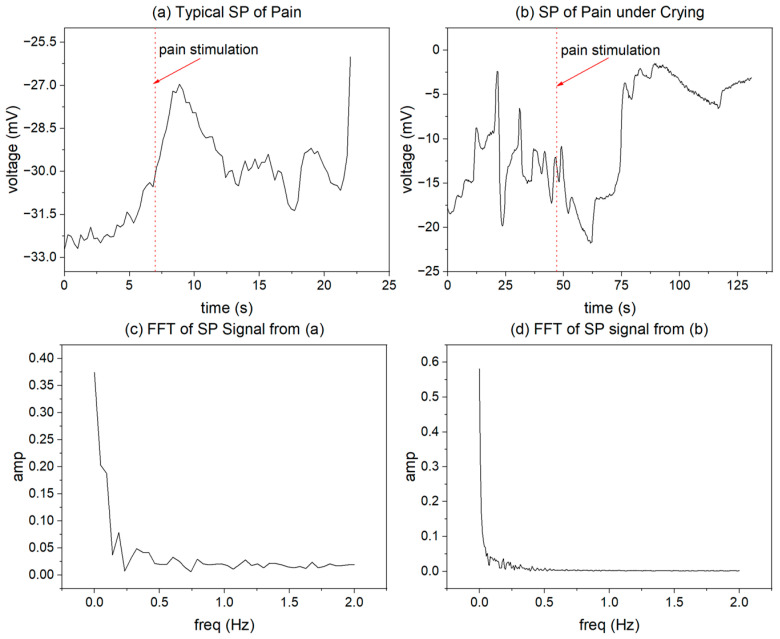
The time-domain plot and Fourier transform results of typical SP signal.

**Figure 3 sensors-23-06815-f003:**
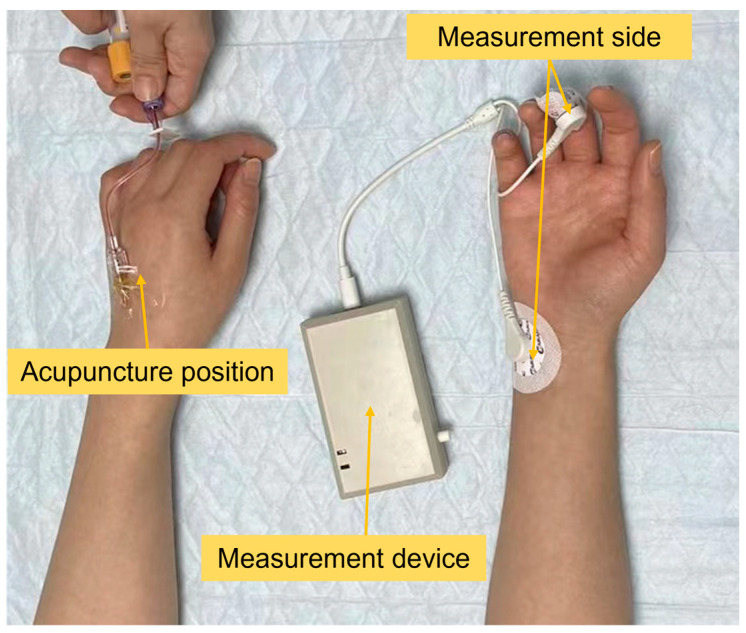
Pain experiment.

**Figure 4 sensors-23-06815-f004:**
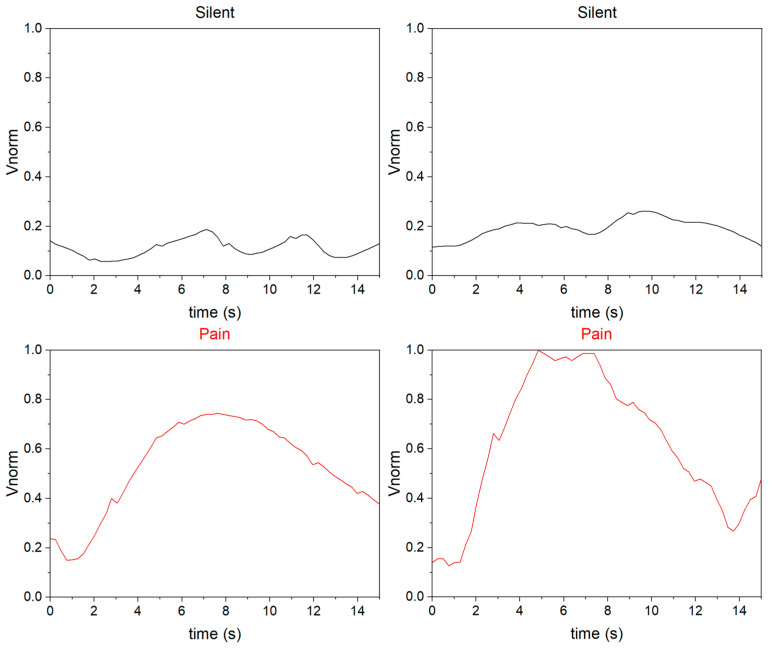
Four typical samples of pain and silent states.

**Figure 5 sensors-23-06815-f005:**
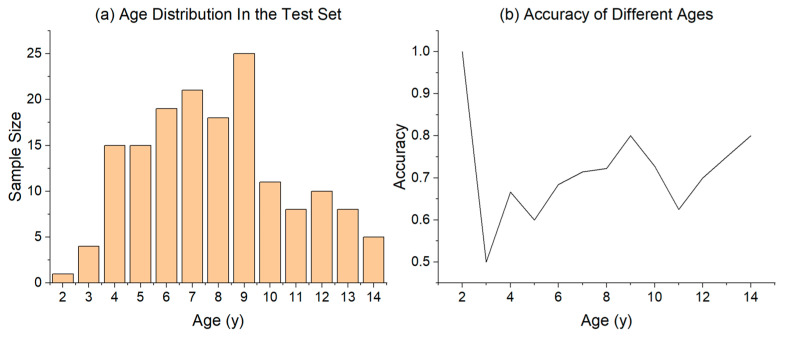
Pain identification results of children of different ages.

**Table 1 sensors-23-06815-t001:** Effective characteristics.

Features	Explanations	Pain Sample Mean (±Standard Deviation)	Silent Sample Mean (±Standard Deviation)
STD	Standard deviation	0.151 (±0.064)	0.104 (±0.047)
Var	Variance	0.027 (±0.023)	0.013 (±0.013)
Diff1_std	Standard deviation of the first difference	0.034 (±0.015)	0.026 (±0.008)
Diff1_abs	Mean of the absolute value of the first difference	0.025 (±0.011)	0.020 (±0.006)
fft_mean	Mean value of the spectrum	0.035 (±0.010)	0.029 (±0.009)
fft_max	The maximum value in the spectrum except for the DC component	0.174 (±0.088)	0.114 (±0.064)
E0	Spectral energy in the 0–0.0625 Hz band	25.321 (±22.443)	13.663 (±12.849)

**Table 2 sensors-23-06815-t002:** Division of training and test sets.

	TagField = 1	TagField = 0
Training set	160	160
Test set	102	58
Total	262	218

**Table 3 sensors-23-06815-t003:** Accuracy of different algorithms and features.

Number of Features	KNN	RF	NN
7	62.50%	70.63%	70.00%
15	60.63%	67.50%	65.00%
25	58.75%	67.50%	64.38%
38	55.00%	68.13%	63.75%

**Table 4 sensors-23-06815-t004:** Effect of sex on the accuracy of pain identification.

Sex	Total Sample Size	Correct Sample Size	Accuracy
Male	87	61	70.11%
Female	73	52	72.60%

## Data Availability

The data presented in this study are available on request from the corresponding author.

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
