# Peer review of "Children’s Pain Identification Based on Skin Potential Signal"

_sensors, 2023, doi:10.3390/s23156815_

Round 1

Reviewer 1 Report

The paper titled "Children Pain Identification Based on Skin Potential Signal " discusses a very interesting topic. The pain that children cannot always be able to express needs an objective measure, which can facilitate the work of the health care provider, and not worsen the condition of the child patient.

The work is presented in detail. All procedures are clear and repeatable.

I would have dedicated a session for age differences among children. Are there differences between children and adolescents? The sample includes subjects up to 18 years of age.

Regarding electrophysiological measurement, one paper demonstrated that PPG (photoplethysmography), an easy-to-use instrument compared to ECG, is a reliable tool for measuring emotion (Rinella et al. 2022. Emotion Recognition: Photoplethysmography and Electrocardiography in Comparison). Therefore, it could also be considered for use in detecting electrophysiological modifications due to pain.

Moreover, in a topic so vast and scientifically relevant, it would be appropriate to expand the references.

Reviewer 2 Report

The authors propose an ML-based model for the quantification of pain identification by skin potential recorded on pediatric subjects.

Major comments:

  1. Table 2.1 shows the feature list. Features 1 and 2, standard deviation, and variance are highly correlated. It should be checked for the performance of the proposed model when excluding one of those features. Also, the standard deviation in the caption of the table should be noted as STD or SD, which will contribute to table visibility.
  2. Abstracts should be rewritten. In Ln 15–16, there is duplicate text such as ‘and lack objectivities. In Ln 20–21, there is duplicate text such as ‘from each sample." Clarify statement conducted experiments with 623 subjects, including 218 pain-free samples and 262 pain samples. This statement is only clear after a complete reading of the manuscripts. A comparison of the results with state-of-the-art methods should be included in the abstract.
  3. Ln 135-138. Clarify statements.
  4. Ln 185: Why do you want to eliminate imbalanced datasets?
  5. Ln 238-242 The font size should be set up.
  6. Ln. 260–262: Clarify statements.
